# Construction of matryoshka nested indecomposable N-replications of Kac-modules of quasi-reductive Lie superalgebras, including the sl(m/n) and osp(2/2n) series

**Jean Thierry-Mieg**[1★], **Peter D. Jarvis**[2,3†] **and Jerome Germoni**[4‡]
**with an appendix by Maria Gorelik**[5∘]

**1** NCBI, National Library of Medicine, National Institute of Health,
8600 Rockville Pike, Bethesda MD20894, U.S.A.
**2** School of Natural Sciences (Mathematics and Physics), University of Tasmania,
Private Bag 37, Hobart, Tasmania 7001, Australia.
**3** Alexander von Humboldt Fellow.
**4** Université Claude Bernard Lyon 1, CNRS UMR 5208,
Institut Camille Jordan, F-69622 Villeurbanne, France.
**5** Department of Mathematics, the Weizmann Institute of Science, Rehovot, Israel.

★ mieg@ncbi.nlm.nih.gov , † peter.jarvis@utas.edu.au ,
‡ germoni@math.univ-lyon1.fr , ∘ maria.gorelik@weizmann.ac.il

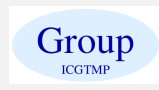
## Abstract

We construct a new class of finite dimensional indecomposable representations of simple superalgebras which may explain, in a natural way, the existence of the heavier elementary particles. In type I Lie superalgebras sl(m/n) and osp(2/2n) , one of the Dynkin weights labeling the finite dimensional irreducible representations is continuous. Taking the derivative, we show how to construct indecomposable representations recursively embedding N copies of the original irreducible representation, coupled by generalized Cabibbo angles, as observed among the three generations of leptons and quarks of the standard model. The construction is then generalized in the appendix to quasi-reductive Lie superalgebras.

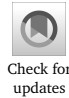

## 1 Introduction

In Kac's complete classification of the simple Lie superalgebras [1, 2], two families contain an even generator $y$ commuting with the even subalgebra, namely the $A(m-1, n-1) = sl(m/n), m \neq n$ and the $C(n+1) = osp(2/2n)$ superalgebras. They admit a single Dynkin diagram with a single odd positive simple root $\beta$ [3].

The even subalgebra, in the corresponding Chevalley basis, has the structure:

$$[h_i, h_j] = 0, \quad [h_i, e_j] = C_{ij} e_j, \quad [h_i, f_j] = -C_{ij} f_j,$$
$$[y, h_i] = [y, e_i] = [y, f_i] = 0, \quad i, j = 1, 2, \ldots, r, \tag{1}$$

where $r$, $h_i$, $e_i$, $f_i$ and $C_{ij}$ denote respectively the rank, the Cartan commuting generators, the raising and the lowering generators associated to the simple roots, and the Cartan matrix of the semisimple even Lie subalgebra $sl(m) \oplus sl(n)$, respectively $sp(2n)$, with rank $r = m+n-2$, respectively $r = n$. The remaining raising (respectively lowering) generators of the even semisimple subalgebra are generated by the iterated commutators of the $e$ (respectively $f$) generators limited by the Serre rule $ad(e_i)(e_j)^{-C_{ij}+1} = 0$. Finally, the additional even generator $y$, that physicists often call the hypercharge, centralizes the even subalgebra. Even in finite dimensional representations, $y$ is not quantized, and as shown below, this is the cornerstone of our new construction of nested indecomposable $N$-replications of an arbitrary Kac module which we propose to call matryoshka representations.

In its odd sector, the superalgebra has $P$ odd raising generators $u_i$ corresponding to the $P$ positive odd roots $\beta_i$ and $P$ odd lowering generators $v_i$ corresponding to the $-\beta_i$, with $P = mn$ for $sl(m/n)$, or $P = 2n$ for $osp(2/2n)$. In both cases, the $u_i$ sit in the irreducible fundamental representation of the even subalgebra. We call $u_1$ the lowest weight vector of the $u_i$ representation; $u_1$ corresponds to the simple positive odd root $\beta = \beta_1$. Reciprocally, we call $v_1$ the highest weight vector of the $v_i$. For our following analysis, the important relations are

$$[y, u_i] = u_i, \quad [y, v_i] = -v_i,$$
$$\{u_i, u_j\} = \{v_i, v_j\} = 0, \tag{2}$$
$$\{u_i, v_j\} = d_{ij}^a \mu_a + ky\, \delta_{ij},$$

where $d_{ij}^a$ and $k$ are constants ($k \neq 0$) and the $\mu_a$ span the even generators of type $(h, e, f)$. That is: the hypercharge $y$ grades the superalgebra, with eigenvalues $(0, \pm 1)$. The $u_i$ anticommute with each other. So do the $v_i$. Finally and most important, the anticommutator of the odd raising operator $u_i$ with the odd lowering operator $v_i$ corresponding to the opposite odd root depends linearly on the hypercharge $y$. In particular, $\{u_1, v_1\} = h_\beta = d_{11}^a h_a + ky$, where $k$ is non zero and $h_\beta$ is the Cartan generator associated to the odd simple root $\beta$. See for example the works of Kac [1,4] or the dictionary on superalgebras by Frappat, Sciarrino and Sorba [5] for details.

## 2 Construction of the Kac modules

Following Kac [4], choose a highest weight vector $\Lambda$ defined as an eigenstate of the Cartan generators $(h_i, y)$, and annihilated by all the raising generators $(e_i, u_j)$. The eigenvalues $a_i$ of the Cartan operators $h_i$ are called the even Dynkin labels. The eigenvalue $b$ of the Cartan operator $h_\beta$ corresponding to the odd simple root is called the odd Dynkin label:

$$h_i \Lambda = a_i \Lambda, \quad \{u_1, v_1\} \Lambda = h_\beta \Lambda = b\Lambda. \tag{3}$$

Construct the corresponding Verma module using the free action on $\Lambda$ of the lowering generators $(f, v)$ modulo the commutation relations of the superalgebra. Since the $v$ anticommute, the polynomials in $(f, v)$ acting on $\Lambda$ are at most of degree $P$ in $v$, and hence the Verma module is graded by the hypercharge $y$ and contains exactly $P$ layers.

Consider the antisymmetrized product $w^-$ of all the odd lowering generators $(v_i, i = 1, 2, ..., P)$. The state $\overline{\Lambda} = w^- \Lambda$ is a highest weight with respect to the even subalgebra $e_i \overline{\Lambda} = 0$. Indeed $e_i$ annihilates $\Lambda$ and each term in the Leibniz development of $[e_i, w^-]$ contains a repetition of one of the $v$ generators, and hence vanishes.

Let $\rho$ be the half supersum of the even and odd positive roots

$$\rho = \rho_0 - \rho_1 = \tfrac{1}{2}(\sum \alpha^+ - \sum \beta^+). \tag{4}$$

Let $w^+$ be the antisymmetrized product of all the $u$ generators. As shown by Kac [4], we have

$$w^+ \overline{\Lambda} = w^+ w^- \Lambda = \pm \prod_i < \Lambda + \rho | \beta_i > \Lambda, \tag{5}$$

where the product iterates over the $P$ positive odd roots $\beta_i$, the sign depends on the relative ordering of $w^+$ and $w^-$ and the bilinear form $< | >$ is a symmetrized version of the Cartan metric. If this product is non-zero, the Verma module is called typical. $\Lambda$ belongs to the orbit of the $\overline{\Lambda}$ and *vice-versa*, hence they both belong to the same irreducible submodule. If the scalar product $< \Lambda + \rho | \beta_i >$ vanishes for one or more odd positive root $\beta_i$, the Verma module is no longer irreducible but only indecomposable since $\Lambda$ is not in the orbit of $\overline{\Lambda}$. It is then called atypical of type $i$ and there exists a state $\omega_i$ with Cartan eigenvalues $\Lambda_i - \beta_i$ which is a sub highest weight annihilated by all the even and odd raising operators $(e, u)$. In the present study, we do not quotient out by this submodule but preserve the indecomposable Verma module construction because we want to preserve the continuity in $b$. Notice that in the $A$ and $C$ superalgebras that we are studying the odd roots are on the light-cone of the Cartan root space: $< \beta_i | \beta_i >= 0$. Therefore, if $\Lambda$ is atypical $i$, the secondary highest weight $\Lambda - \beta_i$ is also atypical $i$.

As in the Lie algebra case, this Verma module is infinite dimensional, because of the acceptable iterated action of the even lowering generators $f$. But as we just discussed, the iterated action of the anticommuting odd lowering operators $v$ saturates at layer $P$.

Let us now recall for completeness the usual procedure to extract a finite dimensional irreducible module from a Lie algebra Verma module. All the states with negative even Dynkin labels which are annihilated by the even raising generators can be quotiented. For example, given a Chevalley basis $(h, e, f)$ for the Lie algebra $sl(2)$ and a Verma module with highest weight $\Lambda$, we have

$$[h, e] = e, \quad [h, f] = -f, \quad [e, f] = 2h,$$
$$h\Lambda = a\Lambda, \quad e\Lambda = 0, \tag{6}$$

hence

$$h f^n \Lambda = (a - 2n) f^n \Lambda, \quad e f^n \Lambda = n(a - n + 1) f^{n-1} \Lambda. \tag{7}$$

If $a$ is a positive integer, the Verma module can be quotiented by the orbit of the state $f^{a+1}\Lambda$, and the equivalence classes form an irreducible module of finite dimension $a + 1$.

Generalizing to a superalgebra, all the even Dynkin labels $a_i$ associated to the Cartan operators $h_i, i = 1, 2, ..., r$ are restricted to non negative integers. We pass to the quotient in each even submodule and define the Kac module as the resulting finite dimensional quotient space. The crucial observation is that the identification of the even sub highest weights $\omega$ requests to solve a set of equations involving the even Dynkin labels $a_i$, but independent of the odd Dynkin weight $b$, which remains non-quantized. For example, in $sl(2/1)$, the state $\omega = (a f v - (a+1) v f)\Lambda$ is an even highest weight [6]. But please remember that we do not quotient out the atypical submodules.

Note that this procedure does not extend to the type II Lie-Kac superalgebras $B(m, n)$, $D(m, n)$, $F(4)$ and $G(3)$, because these algebras contain even generators with hypercharge $y = \pm 2$. For example the generator associated to the lowest weight of the adjoint representation. Indeed, the supplementary root of the (affine) extended Dynkin diagram is even. Thus, the Kac module is finite dimensional if and only if its hidden extended Dynkin label is also a non negative integer. This integrality constraint involves $b$. So the representations of the type II superalgebras are finite dimensional only for quantized values of $b$, see Kac [4] for the original proofs and [7, 8] for examples.

The remaining even highest weights $\Lambda^p_{(...)}$ are spread over the $P$ layers with hypercharge decreasing from $y$ down to $y - P$. On the zeroth layer, we have $\Lambda^0 = \Lambda$, on the first layer we have the $P$ weights $\Lambda^1_i = \Lambda - \beta_i$, on the second layer we have the $P(P-1)/2$ weights $\Lambda^2_{ij} = \Lambda - \beta_i - \beta_j$, $i \neq j$, down to the $P^{th}$ layer $\Lambda^P_{12...P} = \overline{\Lambda}$, each time excluding the even highest weight vectors with negative even Dynkin labels, since they have been quotiented out. For an explicit construction of the matrices of the indecomposable representations of $sl(2/1)$, we refer the reader to our study [6] and references therein.

To conclude, if the Kac module with highest weight $\Lambda$ is typical, it is irreducible. If it is atypical, it is indecomposable. In both cases, its even highest weights are the $\Lambda^p_{(...)}$ and the whole module is given by the even orbits of the $\Lambda^p_{(...)}$ with non negative even Dynkin labels and hypercharge $y - p$.

## 3 On the derivative of the odd raising generators

Consider a finite $D$ dimensional Kac module with highest weight $\Lambda$, typical or atypical, as described in the previous section. Call $a_i$ the even Dynkin labels and $b$ the odd Dynkin label and $y$ the eigenvalue of the hypercharge $Y$ acting on the highest weight state. Notice that $y$ is a linear combination of $b$ and the even Dynkin weights $a_i$. As shown above, in our Chevalley basis, the matrices representing the $(e, f, v)$ generators in the Verma module are by construction independent of $b$, and the matrix representing the hypercharge generator $Y$ can be written as $Y = yI + \alpha^i h_i$, where $I$ is the identity, $h_i$ the even Cartan generators of the semi simple even subalgebra (i.e. excluding $Y$) and the $\alpha^i$ are constants independent of $y$. This remains true in the Kac module because the quotient operations needed to pass to the finite dimensional submodule does not involve $b$. Finally, the matrices representing the odd raising generators $u$ are linear in $b$, i.e. in $y$, because, when we push an odd raising generator $u$ acting from the left through an element of the Kac module, i.e. through a polynomial in $(f, v)$ acting on $\Lambda$, we must contract $u$ with one of the $v$ generators before $u$ touches $\Lambda$.

Now consider the derivatives $u'_i$ of the odd raising $u_i$ matrices

$$u'_i(a) = \partial_y u_i(a, y). \tag{8}$$

Using (2), we derive the anticommutation relations

$$\{u'_i, v_j\} = \partial_y \{u_i, v_j\} = \partial_y (d^a_{ij} \mu_a + k y \delta_{ij}) = k \delta_{ij}, \tag{9}$$

where the $\mu$ matrices span the even generators $(h, e, f)$, and where $\mu(a)$ and $v(a)$ are independent of $y$. Another way of seeing the same results is to compute the $\{u_i(a, y), v_j(a)\}$ anticommutator, divide by $y$ and take the limit when $y$ goes to infinity. Since the matrix elements of the even generators are all bounded when $y$ diverges, except the hypercharge $Y$ with spectrum $y, y - 1, ..., y - P$, we arrive at the same conclusion: the $\{u', v\}$ anticommutator is proportional to the identity on the whole Kac module. Many explicit examples of the matrices $u(a, y), u'(a), v(a)$ can be found in our extensive study of $sl(2/1)$ [6].

This result holds for the Verma modules, for the typical-irreducible Kac modules and for the atypical-indecomposable Kac modules of type I superalgebras, but does not hold for the type II superalgebras or for the irreducible atypical modules of the type I superalgebras because we need continuity in $b$. Indeed, we proved in [9] by a cohomology argument that the fundamental atypical triplet of $sl(2/1)$ cannot be doubled.

The procedure does not hold for the simple superalgebras $psl(n/n)$. It works for $sl(n/n)$, but this superalgebra is not simple because if $m = n$ the $(m/n)$ identity operator $Y$ is supertraceless and generates an invariant 1-dimensional subalgebra that can be quotiented out.

The resulting simple superalgebra $psl(n/n)$ corresponds to the quantized case $y = 0$ and we cannot take the derivative.

# 4 Construction of an indecomposable N-replication of a Kac module

Given a finite $D$ dimensional Kac module, typical or atypical-indecomposable, represented by $D \times D$ matrices $(\mu, y, u, v)$, constructed as above and where $\mu$ collectively denotes the even matrices of type $(h, e, f)$, consider the doubled matrices of dimension $2D \times 2D$:

$$M = \begin{pmatrix} \mu & 0 \\ 0 & \mu \end{pmatrix}, \quad Y = \begin{pmatrix} y & I \\ 0 & y \end{pmatrix}, \quad U = \begin{pmatrix} u & u' \\ 0 & u \end{pmatrix}, \quad V = \begin{pmatrix} v & 0 \\ 0 & v \end{pmatrix}, \tag{10}$$

where we used the $D \times D$ matrices $u'$ constructed in the previous section. By inspection, the matrices $(M, Y, U, V)$ have the same super-commutation relations as the matrices $(\mu, y, u, v)$ and therefore form an indecomposable representation of the same superalgebra of doubled dimension $2D$. This representation cannot be diagonalized since the matrix $Y$ representing the hypercharge cannot be diagonalized because of its block Jordan structure.

The block $u'$ can be rescaled via a change of variables

$$Q = \begin{pmatrix} \lambda & 0 \\ 0 & 1 \end{pmatrix}, \quad Q^{-1} = \begin{pmatrix} 1/\lambda & 0 \\ 0 & 1 \end{pmatrix},$$

$$QUQ^{-1} = \begin{pmatrix} u & \lambda u' \\ 0 & u \end{pmatrix}, \quad QYQ^{-1} = \begin{pmatrix} y & \lambda I \\ 0 & y \end{pmatrix}. \tag{11}$$

Furthermore, we can construct a module of dimension $ND$, for any positive integer $N$ by iterating the previous construction. By changing variables we can then introduce a complex parameter $\lambda$ at each level. For example, for $N = 3$, we can construct

$$M = \begin{pmatrix} \mu & 0 & 0 \\ 0 & \mu & 0 \\ 0 & 0 & \mu \end{pmatrix}, \quad Y = \begin{pmatrix} y & I & 0 \\ 0 & y & I \\ 0 & 0 & y \end{pmatrix}, \quad \tilde{Q}Y\tilde{Q}^{-1} = \begin{pmatrix} y & \lambda_1 I & 0 \\ 0 & y & \lambda_2 I \\ 0 & 0 & y \end{pmatrix},$$

$$U = \begin{pmatrix} u & u' & 0 \\ 0 & u & u' \\ 0 & 0 & u \end{pmatrix}, \quad V = \begin{pmatrix} v & 0 & 0 \\ 0 & v & 0 \\ 0 & 0 & v \end{pmatrix}, \quad \tilde{Q}U\tilde{Q}^{-1} = \begin{pmatrix} u & \lambda_1 u' & 0 \\ 0 & u & \lambda_2 u' \\ 0 & 0 & u \end{pmatrix}. \tag{12}$$

**Matryoshka theorem:** Given any finite dimensional, typical or atypical, Kac module of a type I simple superalgebra, $A(m/n)$, $m \neq n$ or $C(n)$, using the derivative $u'$ of the odd raising generators with respect to the hypercharge $y$ which centralizes the even subalgebra, we can construct an indecomposable representation recursively embedding $N$ replications of the original module.

We propose the name matryoshka because this nested structure strongly resemble the famous Russian dolls.

# 5 Conclusion

Representation theory of Lie algebras and superalgebras involves three increasingly difficult steps: classification, characters and construction. In Lie algebra theory, we can rely on three

major results: all finite dimensional representations of the semisimple Lie algebras are completely reducible, their irreducible components are classified by the Dynkin labels of their highest weight state, their characters are given by the Weyl formula. Nevertheless, the actual construction of the matrices, although known in principle, remains challenging. We only know the matrices in closed analytic form in the case of $sl(2)$.

Finite dimensional simple Lie superalgebras have been classified by Kac [1]. In the present study, we only consider the simple basic classical superalgebras of type 1, $sl(m/n)$, $m \neq n$ and $osp(2, 2n)$ which are characterized by the existence of an even generator, the hypercharge $y$, commuting with the even subalgebra. As for Lie algebras, their irreducible modules can be classified by the Dynkin labels of their highest weight and Kac [4] has discovered in 1977 an elegant generalization of the Weyl formula.

But there are two additional difficulties. First, as found by Kac, the hypercharge $y$ of the finite dimensional modules is not quantized, but for certain discrete values, the Kac module ceases to be irreducible but becomes indecomposable. One can quotient out one or several invariant submodules and the Weyl-Kac formula of the irreducible quotient module is not known in general [10, 11]. Furthermore, there is a rich zoology of finite dimensional indecomposable modules which were progressively discovered by Kac [4], Scheunert [12], Marcu [13, 14], Su [15], and others, culminating in the classification of Germoni [16, 17]. See [6] for an explicit description of the indecomposable $sl(2/1)$ modules.

A particular class, first described by Marcu [14], is of great interest in physics because it has implications for the standard model of leptons and quarks. These particles are well described by $sl(2/1)$ irreducible modules graded by chirality [18–23]. However, experimentally, they appear as a hierarchy of three quasi identical families, for example the muon and the tau behave as heavy electrons. This hierarchical structure has no clear explanation in Lie algebra theory. Furthermore, the three families leak into each other in a subtle way first described by Cabibbo (C) for the strange quarks and generalized to all three families by Kobayashi and Maskawa (KM). In a certain technical sense, the axis of the electroweak interactions is not orthogonal to the axes of the strong interactions, but tilted by small angles, called the CKM angles. As a result the weak interactions are not truly universal because the heavier quarks leak into the lighter quarks. Again, this experimental phenomenon has no explanation in Lie algebra theory precisely because all representations are completely reducible.

Marcu found in 1980 [14] that the fundamental $sl(2/1)$ quartet can be duplicated and triplicated in an indecomposable way. Coquereaux, Haussling, Scheck and coworkers [24–26] have proposed in the 90's to interpret these representations as a description of the CKM mechanism. This raises several questions: is the construction of Marcu limited to three generations, as observed experimentally in the case of the quarks and leptons, or does there exist indecomposable modules involving more layers? Is this property specific of $sl(2/1)$, or is it applicable to other simple Lie-Kac superalgebras?

We have previously partially answered these questions. In [16, 17] the existence of multi generations indecomposable modules is indicated. In [9], we proved, using cohomology, that any Kac module of a type I superalgebra can be duplicated. But these were just proofs of existence.

In the present study, using the derivative of the odd generators relative to the hypercharge, we have shown that any Kac module of a type I Lie-Kac superalgebra $sl(m/n)$, $m \neq n$ and $osp(2/2n)$ can be replicated any desired number of times in an indecomposable way. We have also shown that atypical representations cannot be replicated. We can therefore, in this framework predict the existence of three species of sterile right neutrinos from the observation of the non-zero PMNS (leptonic CKM) mixing angles.

In the appendix presented below (A), we further show that these results are valid for any Kac module $K(L)$ over a quasi-reductive Lie superalgebra $\mathfrak{g}$ of type I. As the reader will no-

tice, the style of this appendix contributed by M.G. is more general and more abstract. We hope that this split/joint presentation will appeal to the wide audience of the G34 conference, equally composed of mathematicians and physicists. The main result is that the "matryoshka N-replication" of the Kac module $K(L)$ has the structure of a module over a Heisenberg superalgebra.

These results are interesting for physics, surprising relative to Lie algebra theory, and very specific as we actually construct the matrices of these "matryoshka" Russian dolls indecomposable modules, in terms of the matrices of the original Kac module, rather than limit our analysis to their existence, classification, or the calculation of their characters.

## Acknowledgments

The authors heartily thank the organizers of the G34 conference

**Author contributions**   JTM, PDJ and JG developed the work on the type 1 superalgebras, MG contributed the generalization to quasi-reductive superalgebras presented in the appendix.

**Funding information**   JTM was supported by the Intramural Research Program of the National Library of Medicine, National Institute of Health. MG was supported by ISF 1957/21 grant.

## A   Appendix: Generalization to quasi-reductive Lie superalgebras

Contributed by Maria Gorelik.

A finite-dimensional Lie superalgebra $\mathfrak{g} = \mathfrak{g}_{\overline{0}} \oplus \mathfrak{g}_{\overline{1}}$ is *quasi-reductive* if $\mathfrak{g}_{\overline{0}}$ is a reductive Lie algebra and $\mathfrak{g}_{\overline{1}}$ is a semisimple $\mathfrak{g}_0$-module. Quasi-reductive Lie superalgebras were introduced in [27, 28]. Simple quasi-reductive Lie superalgebras are classical Lie superalgebras in Kac's classification (see [2]). For other examples and partial classification of quasi-reductive Lie superalgebras, see [27], [28], and [29].

As above, the base field is $\mathbb{C}$ and $\mathfrak{g}$ is a quasi-reductive Lie superalgebra with an even Cartan subalgebra $\mathfrak{h}$. This means that $\mathfrak{h}$ is a Cartan subalgebra of the reductive Lie algebra $\mathfrak{g}_{\overline{0}}$ and that $\mathfrak{g}^{\mathfrak{h}} = \mathfrak{h}$. The only simple quasi-reductive Lie superalgebras which do not satisfy this assumption are $Q$-type superalgebras. We denote by $\mathfrak{h}'$ the center of the reductive Lie algebra $\mathfrak{g}_{\overline{0}}$; one has

$$\mathfrak{h} = \mathfrak{h}' \times \mathfrak{h}'', \quad \text{where } \mathfrak{h}'' := [\mathfrak{g}_{\overline{0}}, \mathfrak{g}_{\overline{0}}] \cap \mathfrak{h}.$$

We identify $(\mathfrak{h}')^*$ with the subspace $(\mathfrak{h}'')^{\perp} = \{\nu \in \mathfrak{h}^* | \ \nu(h) = 0 \ \text{ for any } h \in \mathfrak{h}''\}$. One has

$$\mathfrak{g}_{\overline{0}} = [\mathfrak{g}_{\overline{0}}, \mathfrak{g}_{\overline{0}}] \times \mathfrak{h}'.$$

For an $\mathfrak{h}$-module $N$ we denote by $N_{\nu}$ the generalized weight space corresponding to $\nu \in \mathfrak{h}^*$:

$$N_{\nu} = \{v \in N | \ \forall h \in \mathfrak{h}, \ (h - \nu(h))^s v = 0, \ \text{ for } s >> 0\}.$$

All modules in this section are assumed to be *locally finite over $\mathfrak{h}$ with generalized finite-dimensional weight spaces*: this means that $N = \oplus_{\nu} N_{\nu}$ and $\dim N_{\nu} < \infty$ for all $\nu \in \mathfrak{h}^*$. We set

$$\operatorname{ch} N := \sum_{\nu} \dim N_{\nu} e^{\nu}.$$

A quasi-reductive Lie superalgebra is of *type I* if $\mathfrak{g} = \mathfrak{g}_{-1} \oplus \mathfrak{g}_0 \oplus \mathfrak{g}_1$ is a $\mathbb{Z}$-graded superalgebra. In this case $\mathfrak{g}_{\bar{0}} = \mathfrak{g}_0$ is a reductive Lie algebra, $\mathfrak{g}_{\pm 1}$ are odd commutative subalgebras of $\mathfrak{g}$ and $\mathfrak{h}$ acts diagonally on $\mathfrak{g}_{\pm 1}$. Examples of quasi-reductive Lie superalgebra of type I include $\mathfrak{gl}(m|n)$, $\mathfrak{osp}(2|2n)$, $\mathfrak{p}_n$ and others.

## A.1 Self-extensions of highest weight modules

Let $M$ be a module with the highest weight $\lambda$ (i.e. $M$ is a quotient of $M(\lambda)$), and let $v_\lambda \in M$ be the highest weight vector, i.e. the image of the canonical generator of $M(\lambda)$. Introduce the natural map

$$\Upsilon_M : \text{Ext}^1_{\mathfrak{g}}(M, M) \to \mathfrak{h}^*,$$

as follows. Let $0 \to M \xrightarrow{\phi_1} N \xrightarrow{\phi_2} M \to 0$ be an exact sequence. Let $v := \phi_1(v_\lambda)$ and fix $v' \in N_\lambda$ such that $\phi_2(v') = v_\lambda$. Observe that $v, v'$ is a basis of $N_\lambda$ and so there exists $\mu \in \mathfrak{h}^*$ such that for any $h \in \mathfrak{h}$ one has $h(v') = \lambda(h)v' + \mu(h)v$ (i.e., the representation $\mathfrak{h} \to \text{End}(N_\lambda)$ is $h \mapsto \begin{pmatrix} \lambda(h) & \mu(h) \\ 0 & \lambda(h) \end{pmatrix}$). The map $\Upsilon_M$ assigns $\mu$ to the exact sequence. It is easy to see that $\Upsilon_M : \text{Ext}^1(M, M) \to \mathfrak{h}^*$ is injective.

Notice that if $0 \to M \to N_1 \to M \to 0$ and $0 \to M \to N_2 \to M \to 0$ are two exact sequences then

$$N_1 \cong N_2 \iff \Upsilon_M(N_1) = c \Upsilon_M(N_2), \quad \text{for some } c \in \mathbb{C} \setminus \{0\}. \tag{A.1}$$

If $N$ is an extension of $M$ by $M$ (i.e., $N/M \cong M$) we denote by $\Upsilon_M(N)$ the corresponding one-dimensional subspace of $\mathfrak{h}^*$, i.e. $\Upsilon_M(N) = \mathbb{C}\mu$, where $\mu$ is the image of the exact sequence $0 \to M \to N \to M \to 0$.

### A.1.1

Let $M$ be a finite-dimensional highest weight module. Since $\mathfrak{g}_0$ is reductive, the algebra $\mathfrak{h}''$ acts diagonally on any finite-dimensional $\mathfrak{g}$-module. Therefore the image of $\Upsilon_M$ annihilates $\mathfrak{h}''$, so lies in $(\mathfrak{h}')^*$. In particular, for the image of $\Upsilon_M$ is zero and $\text{Ext}^1(M, M) = 0$ if $\mathfrak{h}' = 0$. In other words, the finite-dimensional highest weight modules do not admit non-splitting self-extensions if $\mathfrak{g}_{\bar{0}}$ is semisimple (for instance, if $\mathfrak{g}$ is a basic classical Lie superalgebra of type II).

## A.2 Kac modules

Let $\mathfrak{g} = \mathfrak{g}_{-1} \oplus \mathfrak{g}_0 \oplus \mathfrak{g}_1$ be a quasi-reductive superalgebra of type I.

The following useful construction appears in several papers including [30]. For a given $\mathfrak{g}_0$-module $M$, we may extend $M$ trivially to a $\mathfrak{g}_0 + \mathfrak{g}_1$-module and introduce the Kac module

$$K(M) := \text{Ind}^{\mathfrak{g}}_{\mathfrak{g}_0 + \mathfrak{g}_1} M.$$

This defines an exact functor $K : \mathfrak{g}_0 - Mod \to \mathfrak{g} - Mod$ which is called *Kac functor*. It is easy to see that $K(M)$ is indecomposable if and only if $M$ is indecomposable.

As $\mathfrak{g}_0$-module we have $K(M) \cong M \otimes \Lambda \mathfrak{g}_{-1}$. Since $\Lambda \mathfrak{g}_{-1}$ is a finite-dimensional module with a diagonal action of $\mathfrak{h}$, $M$ is finite-dimensional (resp., diagonal $\mathfrak{h}$-module) if and only if $K(M)$ is finite-dimensional (resp., diagonal $\mathfrak{h}$-module). Moreover, $M$ is a locally finite $\mathfrak{h}$-module with generalized finite-dimensional weight spaces if and only if $K(M)$ is such a module.

### A.2.1 Self extensions of Kac modules

Take $\nu \in (\mathfrak{h}')^*$ and let $J_n(\nu)$ be the $(n|0)$-dimensional indecomposable $\mathfrak{h}$-module spanned by $v_1, \ldots, v_n$ with the action $h v_i = \nu(h) v_{i+1}$), $h v_n = 0$. ($h$ acts on $V_2(\nu)$ $\begin{pmatrix} 0 & 0 \\ \nu(h) & 0 \end{pmatrix}$). Observe that $h J_n(\nu) = 0$ for all $h \in \mathfrak{h}''$. We view $J_n(\nu)$ as $\mathfrak{g}_0$-module with the zero action of $[\mathfrak{g}_0, \mathfrak{g}_0]$.

For any $\mathfrak{g}_0$-module $L$ the product $J_n(\nu) \otimes L$ is an indecomposable $\mathfrak{g}_0$-module which admits a filtration of length $n$ with the factors isomorphic to $L$. Thus $K(L \otimes J_n(\nu))$ is an indecomposable $\mathfrak{g}$-module which admits a filtration of length $n$ with the factors isomorphic to the Kac module $K(L)$; we denote this module by $K(L; n; \nu)$. In particular, $K(L; 2; \nu)$ is a self-extension of the Kac module $K(L)$.

### A.2.2

Let $L$ be a finite-dimensional highest weight $\mathfrak{g}_0$-module. Then $K(L)$ is a finite-dimensional highest weight $\mathfrak{g}$-module By A.1.1, the image of $\Upsilon_{K(L)}$ lies in $(\mathfrak{h}')^*$. Using the above construction we obtain that the the image of $\Upsilon_M$ is equal to $(\mathfrak{h}')^*$.

### A.2.3

**Lemma.** *If $L$ is a $\mathfrak{g}_0$-module, where $\mathfrak{h}$ acts locally finitely with finite-dimensional generalized weight spaces, then $K(L; n; \nu) \cong K(L; n; \mu)$ if and only if $\nu \in \mathbb{C}^* \mu$.*

*Proof.* If $\nu \in \mathbb{C}^* \mu$, then $J_n(\nu) \cong J_n(\mu)$ and thus $K(L; n; \nu) \cong K(L; n; \mu)$.

Conversely, assume that $K(L; n; \nu) \cong K(L; n; \mu)$. As $\mathfrak{g}_0$-modules

$$K(L; n; \nu) \cong L \otimes J_n(\nu) \otimes \Lambda \mathfrak{g}_{-1} \cong K(L) \otimes J_n(\nu).$$

Thus for any $\lambda \in \mathfrak{h}^*$ we have $K(L; n; \nu)_\lambda \cong K(L)_\lambda \otimes J_n(\nu)$. Take $\lambda$ such that $K(L)_\lambda \neq 0$. Then $K(L; n; \nu) \cong K(L; n; \mu)$ implies

$$K(L)_\lambda \otimes J_n(\nu) \cong K(L)_\lambda \otimes J_n(\mu), \quad \text{as } \mathfrak{h}\text{-modules.} \tag{A.2}$$

We will use the following fact: if $V_1, V_2$ are two modules over a one-dimensional Lie algebra $\mathbb{C}x$ with the minimal polynomials of $x$ on $V_i$ equal to $(x - c_i)_i^k$, then the minimal polynomial of $x$ on $V_1 \otimes V_2$ equals to $(x - (c_1 + c_2))^{k_1 + k_2}$.

Assume that $\nu \notin \mathbb{C}^* \mu$. Then there exists $h \in \mathfrak{h}$ such that $\nu(h) = 0 \neq \mu(h)$. Recall that $h - \lambda(h)$ acts nilpotently on $K(L)_\lambda$, so the minimal polynomial of $h$ on $K(L)_\lambda$ takes the form $(h - \lambda(h))^k$. By above, the minimal polynomial of $h$ on $K(L)_\lambda \otimes J_n(\nu)$ (resp., on $K(L)_\lambda \otimes J_n(\mu)$) is $(h - \lambda(h))^k$ (resp., $(h - \lambda(h))^{k+n}$). Hence (A.2) does not hold: a contradiction. $\square$

## A.3 Action of the Heisenberg superalgebra

Let $\mathfrak{g}$ be a quasi-reductive superalgebra of type I. Retain notation of A.2.1. Let $\iota' : \mathfrak{g}_0 \to \mathfrak{h}'$ be the projection along the decomposition $\mathfrak{g}_0 = [\mathfrak{g}_0, \mathfrak{g}_0] \times \mathfrak{h}'$. We endow the superspace $H = \mathfrak{g}_{-1} \oplus \mathfrak{h}' \oplus \mathfrak{g}_1$ by the structure of Lie superalgebra with the bracket $[-, -]_n$ given by

$$[\mathfrak{g}_1, \mathfrak{g}_1]_n := [\mathfrak{g}_{-1}, \mathfrak{g}_{-1}]_n := 0, \quad [a_-, a_+]_n := \iota'([a_-, a_+]), \quad [h, a_\pm]_n := 0,$$

for all $a_\pm \in \mathfrak{g}_{\pm 1}$ and $h \in \mathfrak{h}'$. Observe that $H$ is a quasi-reductive superalgebra of type I (in fact, $H$ is the direct product an odd Heisenberg superalgebra and a commutative superalgebra). For an $\mathfrak{h}'$-module $M$ we denote by $K_H(M)$ the Kac module for $H$ constructed as in A.2.1.

Let $L$ be a $\mathfrak{g}_0$-module. The construction (10) defines an action of $H$ on a self-extension of a Kac module $K(L)$; we describe this action below.

### A.3.1

Fix $\mu \in (\mathfrak{h}')^*$. Consider a one-parameter family of $n$-dimensional $\mathfrak{g}_0$-modules $V_n(t\nu)$ constructed in A.2.1 (for $t \in \mathbb{R}$). Then $K(L; n; t\nu)$ is a one-parameter family of self-extensions of $K(L)$: this self-extension is splitting if $t = 0$; by (A.1), for $t \neq 0$ all these modules are isomorphic. As a vector space $K(L; n; t\nu))$ is canonically isomorphic to

$$V := L \otimes J_n(\nu) \otimes \Lambda \mathfrak{g}_{-1}.$$

We let $\rho_t : \mathfrak{g} \to \mathrm{End}(V)$ be the representation corresponding to $K(L; n; t\nu)$.

It is easy to see that $\rho_t(u) = \rho_0(u)$ for $u \in [\mathfrak{g}_0, \mathfrak{g}_0] + \mathfrak{g}_{-1}$ and that for $u \in \mathfrak{h}' + \mathfrak{g}_1$ one has $\rho_t(u) = \rho_0(u) + t\rho_t'(u)$ for some $\rho_t'(u) \in \mathrm{End}(V)$.

### A.3.2

**Lemma.** *Define a linear map $\phi : H \to \mathfrak{gl}(V)$ by*

$$\phi(a_-) := \rho_0(a_-), \quad \phi(a_+) := \rho_t'(a_+), \quad \phi(h) := \rho_t'(h),$$

*for $a_\pm \in \mathfrak{g}_\pm$ and $h \in \mathfrak{h}'$. Then $\phi$ is the $H$-representation isomorphic to the Kac module $K_H(L', n, \nu)$ where $L \cong L'$ as a superspace and the action of $H_0 = \mathfrak{h}'$ on $L'$ is trivial.*

*Proof.* Let us check that $\phi$ is a homomorphism of Lie superalgebras. For $a_-, b_- \in \mathfrak{g}_-$ one has $[a_-, b_-]_n = [a_-, b_-] = 0$ and

$$[\phi(a_-), \phi(b_-)] = \rho_0([a_-, b_-]) = 0 = \phi([a_-, b_-]_n).$$

By above, $\frac{\partial^2 \rho_t(a)}{\partial^2 t} = 0$ for any $a \in \mathfrak{g}$. Therefore for any $a, b \in \mathfrak{g}$ one has

$$0 = \frac{\partial^2 \rho_t([a, b])}{\partial^2 t} = [\rho_t'(a), \rho_t'(b)].$$

Taking $a, b \in \mathfrak{g}_1 + \mathfrak{h}'$ we get $0 = [\phi(a), \phi(b)] = \phi([a, b]_n)$.

One has

$$[\rho_t(a), \rho_t'(b)] + [\rho_t'(a), \rho_t(b)] = \rho_t'([a, b]).$$

Using this formula for $a = a_-$ and $b = a_+$ we obtain

$$[\phi(a_-), \phi(a_+)] = [\rho_t(a_-), \rho_t'(a_+)] = \rho_t'([a_-, a_+]) = \rho_t'(\iota'[a_-, a_+]) = \phi([a_-, a_+]_n).$$

Finally, taking $h \in \mathfrak{h}'$ and $a_- \in \mathfrak{g}_-$ we get

$$[\phi(h), \phi(a_-)] = [\rho_t'(h), \rho_t(a_-)] = \rho_t'([h, a_-]) = 0,$$

so $[\phi(h), \phi(a_-)] = \phi([h, a_-]_n)$. Hence $\phi$ is a homomorphism, so $\phi$ defines a representation of $H$. Denote the corresponding $H$-module by $N$. Let us check that the linear isomorphism $L \xrightarrow{\sim} L'$ induces the $H$-module isomorphism $N \xrightarrow{\sim} K_H(L', n, \nu)$.

Since $K(L, n, t\nu)$ is a free $\mathfrak{g}_{-1}$-module generated by the subspace $L \otimes V_n(t\nu)$, $N$ is a free $\mathfrak{g}_{-1}$-module generated by the subspace $L' \otimes V_n(t\nu)$. For any $v \in L' \otimes V_n(t\nu)$ one has $\rho_t(a_+)(v) = 0$, so $\phi(a_+)(v) = 0$. Take $h \in \mathfrak{h}'$. Let $v_1, \ldots, v_n$ be the standard basis of $J_n(\nu)$ (see A.2.1). For $w \in L$ we have

$$\rho_t(h)(w \otimes v_i) = \rho_0(h)(w) \otimes v_i + t\nu(h)w \otimes v_{i+1},$$

so $\phi(h)(w \otimes v_i) = \nu(h)w \otimes v_{i+1}$. Hence $N = K_H(L', n, \nu)$ as required. $\qquad\square$

## A.4 Another construction of the action of the Heisenberg superalgebra

A natural question is to find a more "natural" construction for Heisenberg superalgebra and its action. This can be done as follows.

### A.4.1

Let $\mathfrak{t}$ be any Lie superalgebra. We introduce the increasing filtration by $\mathcal{F}^0(\mathfrak{t}) = 0$, $\mathcal{F}^1(\mathfrak{t}) := \mathfrak{t}_{\bar{1}}$ and $\mathcal{F}^2(\mathfrak{t}) := \mathfrak{t}$.

The associated graded Lie superalgebra $\tilde{H} := \mathrm{gr}_{\mathcal{F}}(\mathfrak{t})$ is naturally isomorphic to $\mathfrak{t}$ as a vector superspace; denoting this linear isomorphism by $\iota : \mathfrak{t} \to \tilde{H}$ we obtain the following formulae for the bracket on $\tilde{H}$:

$$[\tilde{H}_{\bar{0}}, \tilde{H}] = 0, \quad [a, b] := \iota([\iota^{-1}(a), \iota^{-1}(b)]), \quad \text{if } a, b \in \tilde{H}_{\bar{1}}.$$

If $\dim \mathfrak{t} < \infty$, then $\tilde{H}$ is quasi-reductive. If $\mathfrak{t}$ is $\mathbb{Z}$-graded and finite-dimensional, then $\tilde{H}$ is quasi-reductive of type I ($\tilde{H}$ is the direct product an odd Heisenberg superalgebra and a commutative superalgebra).

If $N$ is a $\mathfrak{t}$-module generated by a subspace $N'$ we can define a compatible increasing filtration on $N$ by setting $\mathcal{F}^0(N) = N'$ and $\mathcal{F}^i(N) = \mathcal{F}^i(\mathcal{U}(\mathfrak{t}))N'$. The associated graded module $\mathrm{gr}_{\mathcal{F}}(N)$ has a structure of $\tilde{H}$-module.

### A.4.2 Application to A.3

Retain notation of A.3. Let $\mathfrak{t} := \mathfrak{g}$ be quasi-reductive of type I. Identify $\tilde{H}_{\bar{0}}$ with $\mathfrak{g}_{\bar{0}} = \mathfrak{g}_0$. Since $\tilde{H}_{\bar{0}}$ lies in the center of $\tilde{H}$, any subspace of $\mathfrak{g}_0$ is an ideal of $\tilde{H}$. It is easy to see that $\tilde{H}/[\mathfrak{g}_0, \mathfrak{g}_0]$ is isomorphic to $H$ constructed in A.3.

Set $N := K(L, n, \nu)$. Fix $h \in \mathfrak{h}'$ such that $\nu(h) = 1$. Recall that the $h$ acts on $J_n(\nu)$ by a Jordan cell: $J_n(\nu)$ is spanned by $v_1, hv_2, h^2 v_1, \ldots, h^{n-1} v$. $L \otimes J_n(\nu)$ is spanned by $L \otimes v_1, h(L \otimes v_1), \ldots, h^{n-1}(L \otimes v_1)$, so $N' := L \otimes v_1$ generates $N$ over $\mathfrak{g}$. Define the increasing filtration on $N$ as above. The associated graded module $\mathrm{gr}_{\mathcal{F}}(N)$ is a $\tilde{H}$-module. We have $[\mathfrak{g}_0, \mathfrak{g}_0](\mathcal{F}^i(N)) = \mathcal{F}^i(N)$ for each $i$, so $[\mathfrak{g}_0, \mathfrak{g}_0]$ annihilates $\mathrm{gr}_{\mathcal{F}}(N)$. Hence $\mathrm{gr}_{\mathcal{F}}(N)$ is an $H$-module. It is not hard to see that this module is isomorphic to the $H$-module constructed in Lemma A.3.2.

Conclusion: the matryoshka N-replication of the Kac module $K(L)$ has the structure of a module over a Heisenberg superalgebra.

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
