# Peer review of "Construction of matryoshka nested indecomposable N-replications of Kac-modules of quasi-reductive Lie superalgebras, including the sl(m/n) and osp(2/2n) series"

_SciPost Physics Proceedings, doi:SciPost Phys. Proc. 14, 045 (2023)_

## Round 2 · Referee Report · Anonymous (Referee 1) · 2023-1-25

**Report** on the

**Paper** : "*Construction of matryoshka nested indecomposable $N$-replications of Kac-modules of quasi-reductive Lie superalgebras, including the $sl(m|n)$, $osp(2|2n)$ series*"

**Authors** : Jean Thierry-Mieg, Peter Jarvis, and Jerome Germoni (appendix by Maria Gorelik)

### Mathematical abstract :

The present paper exploits a crucial feature of type I Lie superalgebras $A(m-1, n-1) = sl(m|n)$ ($m \neq n$; see minor observation 1 further below) and $C(n+1) = osp(2|2n)$ : they contain an even generator $y$, which physicists usually name *hypercharge*, which commutes with the even subalgebra, and gives rise to a 3-grading of the whole superalgebra, correspondingly admitting a single Dynkin diagram with a single odd positive simple root $\beta$.

Within such Lie superalgebras, the anti-commutator between odd raising and lowering operators reads
$$\{u_i, v_j\} = d_{ij}^a \mu_a + ky\delta_{ij},$$
where $k \neq 0$ and $d_{ij}^a$ are constants (generally depending on the superalgebras), and the $\mu_a$'s span the even generators of type $h, e, f$ in the 0-graded (even) part of the superalgebra.

Considering the extraction of a finite dimensional irreducible Kac module from the Verma module of such type I Lie superalgebras, the crucial observation made by the authors is that the identification of the even sub highest weights $\omega$ requests to solve a set of equations involving the even[1] Dynkin labels $a_i$, but *independent* of the odd Dynkin weight $b$, which thus remains *non-quantized*. This fact does *not* extend to the type II Lie-Kac superalgebras $B(m,n)$, $D(m,n)$, $F(4)$ and $G(3)$, because these algebras contain even generators with hypercharge eigenvalues $\pm 2$, i.e. they are not only 3-graded under the generator $y$.

This yields to the conclusion that $u_i = u_i(a, y)$ and $v_j = v_j(a)$ only, such that one can define
$$u_i'(a) := \partial_y u_i(a, y),$$
and thus compute (see minor observation 2 below)
$$\{u_i', v_j\} = \partial_y \{u_i, v_j\} = \partial_y \left(d_{ij}^a \mu_a + ky\delta_{ij}\right) = k\delta_{ij},$$
a result which holds for the Verma modules, for the typical-irreducible Kac modules and for the atypical-indecomposable Kac modules of type I superalgebras, but does *not* hold for the type II superalgebras or for the irreducible atypical modules of the type I superalgebras, because in such cases $b$ is quantized and the corresponding differentiation cannot be performed.

The main result of the paper is the

*Matryoshka theorem*: Given any finite dimensional, typical or atypical, Kac module of a type I superalgebra, $A(m|n)$ or $C(n)$, using the derivative $u'$ of the odd raising generators with respect to the hypercharge $y$ which centralizes the even subalgebra, one can construct an indecomposable representation recursively embedding $N$ replications of the original module, $N \in \mathbb{N}$.

In the appendix, due to M. Gorelik, the above theorem is extended to any Kac module over a quasi-reductive Lie superalgebra $g$ of type I : the "matryoshka $N$-replication" of the Kac module has the structure of a module over a Heisenberg superalgebra.
* * *
[1]In this context, as usual, the eigenvalues $a_i$ of the Cartan operators $h_i$ are called the *even* Dynkin labels, whereas $b$ is the eigenvalue of the Cartan operator $h_\beta$ corresponding to the odd simple root $\beta$, and it is called the *odd* Dynkin label.

**Physical considerations :**

At least for $sl(2|1)$, the result of the paper is of great interest in physics, because it has implications for the leptons and quarks, as they are described by the Standard Model of elementary particles : in fact, leptons and quarks are well described by $sl(2|1)$ irreducible modules graded by chirality. However, the hierarchical structure of three quasi-identical families of leptons and quarks has no clear explanation in Lie algebra theory, nor has the existence of the CKM matrix. In 1980 it was found that the fundamental $sl(2|1)$ quartet can be duplicated and triplicated in an indecomposable way, and this was later proposed to be a description of the CKM mechanism. So, the natural question arises out whether such a replication *is limited to three generations*, as observed experimentally in the case of quarks and leptons, or whether there exist indecomposable modules involving more layers. Moreover, one could ask whether this property is specific of $sl(2|1)$, or if it is applicable to other simple Lie-Kac superalgebras. The authors had already provided a proof of existence that any Kac module of a type I superalgebras can be duplicated, but in the present paper they have explicitly shown that any Kac module of a type I Lie-Kac superalgebra $sl(m|n)$ and $osp(2|2n)$ can be replicated *any desired number of times* in an indecomposable way.

**Final judgement :**

The paper is very well written, and actually it splits into two parts : the main text is more suitable for a mathematical physicists' audience, whereas the appendix is more abstract and better suited for mathematicians. Besides the elegance of the result and its intrinsical mathematical interest, the papers is of great interest in relation to possible applications to the physics of the Standard Model of elementary particles, as resulting from the above physical considerations, which are reported in the Conclusion by the authors themselves.

Alas, yet *no answer is provided* to the question whether the triplication ($N = 3$) into three hierarchical families of quarks and leptons, as observed in Nature and unexplained in the Standard Model, has any reason to stand out as a kind of "sweet spot" in the matryoshka $N$-replication of the Kac modules of the $sl(2|1)$ Lie superalgebra.

All in all, I recommend the present paper for publication, provided that the authors address the following two
**Minor observations :**

1. what about $A(n-1, n-1) = sl(n|n)/\mathcal{Z}$ ? This is also a Lie superalgebra of type I, and its even part is $sl(n) \oplus sl(n)$, so in this case there is no even generator commuting with the even subalgebra.

2. Eq. (3.2) seems to contain a typo after the second $=$. Indeed, shouldn't $\partial_b$ be replaced by $\partial_y$ ?

---

## Round 3 · Author Response

We are grateful to the referee for noticing a typo in equation 3.2 and for suggesting that we should consider separately the case sl(n/n) and psl(n/n). There are no other significant change to the manuscript.

---

## Round 3 · List of Changes

Following the suggestions of the referee, we edited a typo in equation 3.2 and specified in the introduction, the statement of the theorem and the conclusion that the construction applies to the simple superalgebras sl(m/n), only when m \neq n. At the end of section 3, we further state why the construction does not apply to the case psl(n/n) . We also slightly edited the abstract.

---

## Editorial Decision

published